# *Where* AND *Why* IN IMAGE FORGERY: A BENCHMARK FOR JOINT LOCALIZATION AND EXPLANATION

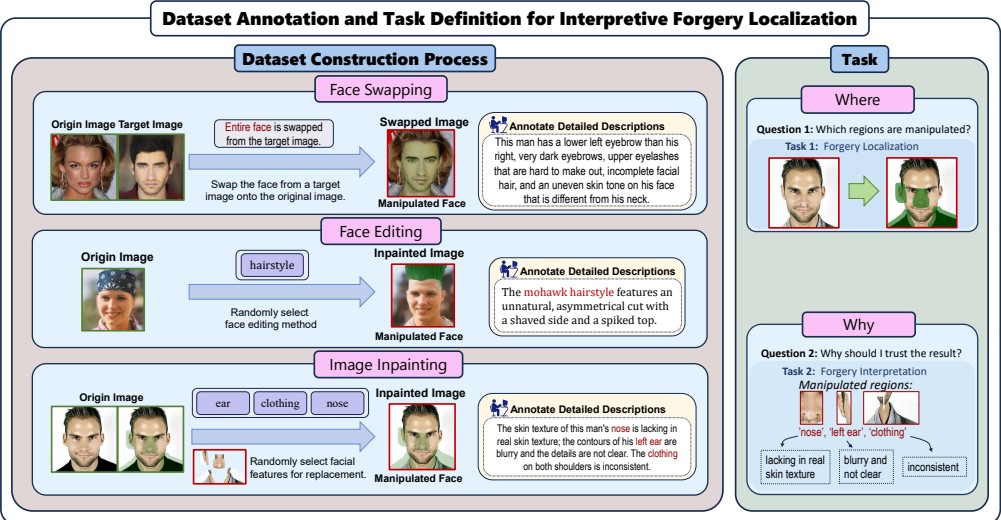

Figure 1: An overview of our proposed benchmark, illustrating the dataset construction process and the joint task definition. The left panel shows the 3 typical manipulation paradigms used for data generation, *i.e.*, Face Swapping, Face Editing, and Image Inpainting. The right panel defines the task of Joint Localization and Explanation, which requires models to answer both "where" a forgery is (Localization) and "why" it is a forgery (Explanation).

## ABSTRACT

Existing facial forgery detection methods typically focus on binary classification or pixel-level localization, providing little semantic insight into the nature of the manipulation. To address this, we introduce Forgery Attribution Report Generation, a new multimodal task that jointly localizes forged regions ("Where") and generates natural language explanations grounded in the editing process ("Why"). This dual-focus approach goes beyond traditional forensics, providing a comprehensive understanding of the manipulation. To enable research in this domain, we present **M**ulti-**M**odal **T**amper **T**racing (**MMTT**), a large-scale dataset of 152,217 samples, each with a process-derived ground-truth mask and a human-authored textual description, ensuring high annotation precision and linguistic richness. We further propose ForgeryTalker, a unified end-to-end framework that integrates vision and language via a shared encoder (image encoder + Q-former) and dual decoders for mask and text generation, enabling coherent cross-modal reasoning. Experiments show that ForgeryTalker achieves competitive performance on both report generation and forgery localization subtasks, *i.e.*, 59.3 CIDEr and 73.67 IoU, respectively, establishing a baseline for explainable multimedia forensics. Dataset and code will be released to foster future research.

## 1 INTRODUCTION

The emergence of advanced generative models, particularly diffusion models (Ho et al., 2020; Song et al., 2020; Zhang et al., 2025), has significantly enhanced the sophistication and realism of im-

Table 1: Comparison of Face Manipulation Datasets. Our MMTT dataset is highlighted and provides rich text annotations for the *"why"* problem, a unique feature among existing resources.

| Dataset | Tasks | Modality | Source Samples | Unique Forgeries | Manipulation Type | GT Type |
|---|---|---|---|---|---|---|
| FaceForensics++ (Rossler et al., 2019) | Class. / Seg. | Video | 1,000 | 4,000 | Multi-Face Mods | Label + Mask |
| Celeb-DF (Li et al., 2020) | Classification | Video | 590 | 5,639 | DeepFake | Image Label |
| DeeperForensics-1.0 (Jiang et al., 2020) | Classification | Video | 50,000 | 10,000 | GAN | Image Label |
| DFDC (Dolhansky et al., 2020) | Classification | Video | 23,654 | 104,500 | DeepFake | Image Label |
| FaceShifter (Li et al., 2019) | Classification | Video | N/A | 5,000 | GAN | Image Label |
| ForgeryNet (He et al., 2021) | Class. / Seg. | Image, Video | 116,321 | 221,247 | DeepFake, GAN | Label + Mask |
| OpenForensics (Le et al., 2021) | Detection / Seg. | Image, Video | 45,473 | 70,325 | GAN, Inpainting | BBox, Mask |
| DF40 (Yan et al., 2024) | Class. / Seg. | Image, Video | N/A | > 1,000,000 | Multi-Face Mods | Label + Mask |
| DiffusionFace (Chen et al., 2024) | Generation | Image | N/A | 50,000 | Diffusion | Image Label |
| GenFace (Zhang et al., 2024a) | Generation | Image | 10,000 | 10,000 | GAN, Inpainting | Mask |
| **MMTT (Ours)** | **Seg. / Caption** | **Text, Image** | **100,000** | **152,217** | **Face Swap, Inpainting, Attribute Edit** | **Text + Mask** |

age generation techniques, making them increasingly difficult to detect. While these techniques have shown immense potential in creative fields such as digital art and film production (Dhariwal & Nichol, 2021; Liu et al., 2025a), they have also raised profound concerns about their misuse in malicious contexts, including misinformation campaigns and privacy violations (Rana et al., 2022; Liu et al., 2023; Zhu et al., 2025b; Liu et al., 2025b), especially the manipulation of facial images. Given these threats, DeepFake detection techniques have garnered significant attention and have rapidly evolved in recent years. Recent studies are shifting from simple real-fake detection to fine-grained forgery region localization to address the growing complexity of modern forgery techniques (Verdoliva, 2020; Rossler et al., 2019; Wu et al., 2023; Yu et al., 2021).

Unlike binary classification methods, which merely determine whether an image is fake or real, forgery localization segments the exact areas that have been tampered with (Verdoliva, 2020), aiming to give the reason behind a forgery determination. However, binary masks, which merely highlight tampered pixels, provide limited insights into the rationale behind the model's predictions (Rossler et al., 2019). Furthermore, these masks fail to differentiate between subtle and more significant alterations, treating all manipulated pixels equally, which often obscures the most critical areas that warrant closer scrutiny. Meanwhile, modern forgeries are often visually indistinguishable from real images. This makes it challenging for even human reviewers to identify tampered regions. For example, slight modifications in facial features, such as subtle distortions of the eyes or lips, are often overlooked in existing works, providing human observers with insufficient information to recognize the most anomalous regions and trust the recognition results.

To address these limitations, we propose and establish a benchmark for a new multimodal task: Joint Forgery Localization and Explanation. The goal is to answer both "where" a forgery is located and "why" it is identified as a forgery. To achieve this, models must concurrently generate a pixel-level localization mask and a natural language report detailing the manipulation artifacts. To catalyze research on this task, we construct and release the **M**ulti-**M**odal **T**amper **T**racing (**MMTT**) dataset, the first large-scale benchmark for this purpose. MMTT contains 152,217 samples, each comprising a forged image, its high-precision process-derived ground-truth mask, and a meticulously crafted human-authored textual description. Building on this benchmark, we propose **ForgeryTalker**, a novel baseline model that offers a unified, end-to-end solution to this joint task. At its core, ForgeryTalker is designed to integrate vision and language reasoning within a single, cohesive architecture. It utilizes a shared encoder to learn a common, forgery-aware representation from the input image, forcing a deep fusion of visual and semantic features. This shared understanding is then processed by specialized dual decoders to generate both the pixel-level localization mask and the natural language report. This integrated design is crucial, enabling the model to produce a coherent attribution report where the textual explanation is semantically grounded in the visual evidence highlighted by the mask, thereby directly addressing the shortcomings of traditional methods. Our primary contributions are summarized as follows:

- **A New Task and Dataset.** We introduce *Forgery Attribution Report Generation*, a new multimodal task that jointly addresses the "Where" and "Why" of image forgery by combining pixel-level localization and natural language explanation. To enable this research, we present **M**ulti-**M**odal **T**amper **T**racing (**MMTT**), a large-scale dataset with 152,217 samples, each featuring a process-derived ground-truth mask and a human-authored textual description. This ensures both high precision in localization and semantic richness in explanation.

- **A Unified and Effective Baseline.** We propose ForgeryTalker, a unified framework that jointly performs forgery localization and report generation. It is designed to facilitate coherent cross-

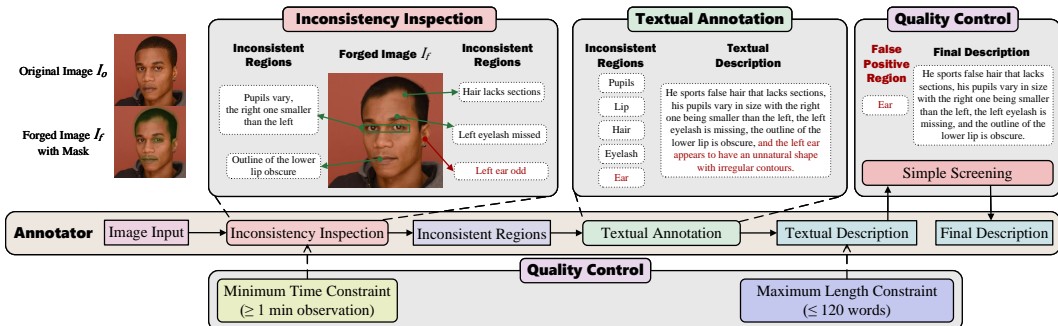

Figure 2: The manual annotation pipeline for the MMTT dataset. Here we show the key stages from inconsistency inspection to textual description and final quality control.

modal reasoning through a shared encoder (image encoder + Q-former) and dual decoders (mask decoder and a Large Language Model).

- **Comprehensive Benchmarking.** We conduct extensive experiments on the proposed dataset in both report generation and forgery localization tasks. It validates that, albeit simple, our baseline achieves competitive performance, *i.e.*, 59.3 CIDEr and 73.67 IoU, and the complementary effects between two tasks.

## 2 MULTI-MODAL TAMPER TRACING DATASET

As a comparison with other major forgery datasets in Table 1 highlights, the proposed MMTT is the first to provide detailed textual explanations alongside forgery masks. The dataset contains **152,217** samples distributed across four manipulation paradigms, with diffusion-based inpainting and face swapping being the most prevalent (Figure 3a). Our statistical analysis reveals several key properties: (1) Eyebrows, eyes, and lips are the most common targets for manipulation across all localized editing methods (Figure 3b). (2) A significant portion of images feature multiple alterations, with 2-5 concurrent modifications being common (Figure 3c). (3) The textual annotations form a rich corpus of over 4 million words, with an average description length of **27.4 words**. The content of these descriptions aligns closely with the visual forgeries, frequently referencing the manipulated facial parts (Figure 3d). As shown in Figure 1, our MMTT dataset provides two complementary types of annotations: *binary forgery masks* in Section 2.1 and *forgery analysis text* in Section 2.2. The forgery analysis text primarily delivers diagnostic summaries of facial images, while the binary masks serve as auxiliary clues, highlighting localized forgery artifacts.

### 2.1 FORGERY GENERATION

To construct a challenging and diverse dataset, we simulate forgery threats using three distinct manipulation paradigms. For each, we employed state-of-the-art models and developed specific procedures to programmatically generate forged images $I_f$ and their corresponding pixel-perfect ground-truth masks $M$.

**Source Image Collection.** We first construct the MMTT dataset from 100,000 high-quality facial images, comprising 30,000 images from CelebAMask-HQ (Zhu et al., 2022) and 70,000 images from Flickr-Faces-HQ (FFHQ) (Karras et al., 2019). All images are resized to $512 \times 512$ pixels, which serve as the primary source for subsequent forgery manipulations.

**Face Swapping.** We used the GAN-based E4S (Abou Akar et al., 2024) model to swap faces between randomly paired images from our source datasets. Crucially, the E4S model automatically generates a precise binary mask $M$ during this process, which directly serves as the ground-truth annotation for the manipulated region in the final forged image $I_f$.

**Face Editing.** We performed semantic alterations using GAN-inversion models StyleCLIP (Patashnik et al., 2021) and HFGI (Wang et al., 2022). The transformation is applied to an input image $I$ to produce the forged image $I_f = \mathcal{E}_{\text{model}}(I, a)$, where $\mathcal{E}$ is the editing function guided by attribute $a$, and model $\in \{\text{StyleCLIP}, \text{HFGI}\}$. The corresponding ground-truth mask, $M_{\text{final}}$, is constructed by taking the union of any pre-existing mask ($M_{\text{prev}}$) and a new semantic mask ($M_{\text{semantic}}$) generated via a face parsing model (Yu et al., 2018), formulated as $M_{\text{final}} = M_{\text{prev}} \cup M_{\text{semantic}}$.

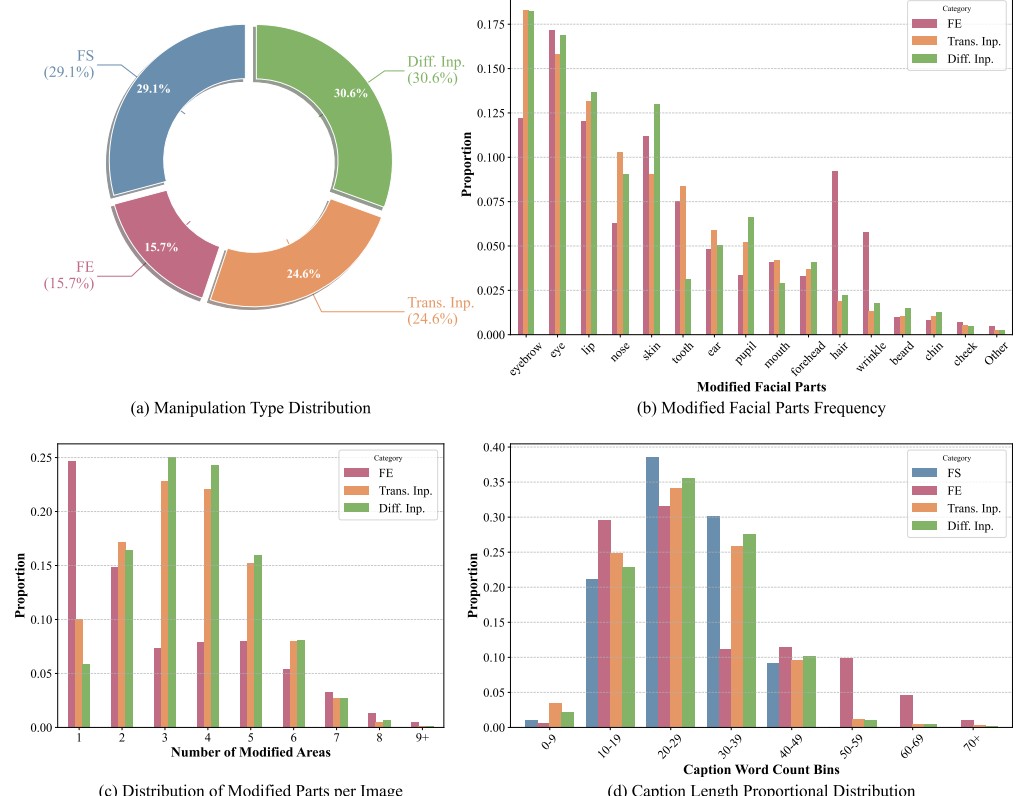

(a) Manipulation Type Distribution

(b) Modified Facial Parts Frequency

(c) Distribution of Modified Parts per Image

(d) Caption Length Proportional Distribution

Figure 3: Statistical overview of the MMTT dataset and its four manipulation types: Face Swapping (FS), Face Editing (FE), Transformer-based Inpainting (Trans. Inp.), and Diffusion-based Inpainting (Diff. Inp.). The figure shows: (a) the proportional distribution of these types; (b) the modification frequency of facial parts for localized edits (excluding FS, which alters the entire face); (c) the number of concurrently modified parts per image; and (d) the word count distribution of the corresponding textual descriptions.

**Image Inpainting.** We generated localized forgeries using both transformer-based (MAT (Li et al., 2022)) and diffusion-based (SDXL (Podell et al., 2023)) models. The required input masks ($M$) were created by programmatically selecting and merging facial component segments. The final inpainted image $I_f$ is produced by composing the original image $I$ with the model's output $I_g^{\text{model}}$ using the mask $M$: $I_f = (1 - M) \cdot I + M \cdot I_g^{\text{model}}$, where model $\in \{\text{MAT}, \text{SDXL}\}$.

## 2.2 Diagnosis Text Annotation

**Annotation Methodology.** To ensure high-quality annotations, we develop a structured pipeline (see Figure 2) where a team of expert annotators receives specific guidelines. Guided by a ground-truth mask for each image pair, annotators are instructed to describe visual inconsistencies or artifacts exclusively within the manipulated regions. They focus only on unnatural or poorly-integrated features, omitting descriptions of authentic areas, and compose self-contained descriptions that do not reference the original image. Each description is limited to a maximum of 120 words.

**Annotation Process.** Our annotation process involves 30 trained annotators who follow a three-step procedure. First, annotators receive an original-forgery image pair $(I_o, I_f)$ with its corresponding ground-truth mask $M$. They then inspect the images for inconsistencies within the masked facial regions, such as unnatural textures or asymmetries. Finally, they compose a detailed textual description $T$, explaining the specific nature of the alteration. This process culminates in a triplet $p = (I_f, M, T)$.

**Annotation Quality Assurance.** To ensure description reliability, we implement a series of quality control measures. We enforce a minimum observation time of one minute per image pair to ensure

Figure 4: Illustration of our ForgeryTalker framework. The training pipeline has two stages. In the Forgery-aware Pretraining Stage, the Q-former, Mask Decoder, and Language Model are jointly optimized with MLM, language modeling, segmentation, and contrastive losses to build multimodal representations. In the Explanation Generation Stage, the FPN is trained with BCE and Dice losses for region classification and then frozen while the Q-former and Mask Decoder are fine-tuned for improved forgery localization and explanation. Finally, the multimodal features are fed to a Large Language Model to generate explanatory reports.

thorough examination. Textual descriptions referencing regions outside the ground-truth mask are automatically flagged for review to prevent false positives and maintain annotation accuracy.

## 3 FORGERYTALKER

The architecture of our baseline model, **ForgeryTalker**, extends InstructBlip (Dai et al., 2023) and is structured around a shared encoder and dual decoders. The shared encoder, consisting of a Vision Transformer and a Q-Former, processes the tampered image $I$ to extract multimodal features. Guided by prompts from an integrated Forgery Prompter Network (FPN), these features are then passed to two decoders: a Mask Decoder for forgery localization and a Large Language Model (LLM) that generates the final attribution report. As shown in Figure 4, training proceeds in two stages. In the **Forgery-aware Pretraining Stage**, we jointly optimize the core modules using a weighted combination of losses to build forgery-sensitive multimodal representations. In the subsequent **Attribution Report Generation Stage**, we first train the FPN to generate accurate region prompts. Then, with the FPN fixed, we fine-tune the mask decoder and Q-Former using segmentation and language modeling losses to improve forgery localization and the final attribution report.

### 3.1 FORGERY-AWARE PRETRAINING

The goal of our forgery-aware pretraining stage is to learn robust multimodal representations that are sensitive to manipulation artifacts. Given an image $I$ and its corresponding ground-truth explanation text $T$, we jointly optimize the core modules of our model using four distinct training objectives. The image $I$ is first processed by a frozen visual encoder to yield embeddings $E_I$, which serve as input alongside the text $T$ for the following loss functions:

**Masked Language Modeling ($\mathcal{L}_{mlm}$):** The text $T$ is tokenized into $\tilde{T}$. Before feeding $\tilde{T}$ into the Q-Former, a subset of region-related tokens (*e.g.*, "ear", "eye", etc.) $\mathcal{M}$ is masked, and the masked token results in $\tilde{T}_{\backslash\mathcal{M}}$. Along with the learned query tokens $Q$ and image embeddings $E_I$, the Q-Former predicts the masked tokens. The loss is computed as:

$$\mathcal{L}_{mlm} = -\sum_{t \in \mathcal{M}} \log P(t \mid I, \tilde{T}_{\backslash\mathcal{M}}). \tag{1}$$

**Language Modeling ($\mathcal{L}_{lm}$):** The Q-Former output is projected and fed to a T5-based decoder (Chung et al., 2022) that generates the explanatory text $\hat{T}$ with the length of $L_{\hat{T}}$. The generated

explanation is compared token-by-token with the ground truth via cross-entropy loss:

$$\mathcal{L}_{lm} = -\sum_{k=1}^{L_{\hat{T}}} \log P\left(\hat{T}_k \mid I, \hat{T}_0, \ldots, \hat{T}_{k-1}\right), \tag{2}$$

**Forgery Localization ($\mathcal{L}_{seg}$):** The non-[CLS] tokens of $E_I$ are seamlessly fused with the text $T$ via cross-attention. The mask decoder predicts a forgery mask $\hat{M}$ with the height $H$ and width $W$, which is compared to the ground-truth mask $M$ using pixel-wise cross-entropy loss:

$$\mathcal{L}_{seg} = -\frac{1}{HW} \sum_{i=1}^{H} \sum_{j=1}^{W} \left[ M_{ij} \log \hat{M}_{ij} + (1 - M_{ij}) \log(1 - \hat{M}_{ij}) \right], \tag{3}$$

where $M_{ij} = 1$ if the $(i, j)$ pixel is manipulated, 0 otherwise. **Cross-model Alignment Learning** ($\mathcal{L}_{con}$)**:** To align modalities, we pull the global image feature $v$ (from the [CLS] token) closer to the mean-pooled text feature $t$ with contrastive loss as:

$$\mathcal{L}_{con} = -\frac{1}{N} \sum_{i=1}^{N} \log \frac{\exp\left(\mathrm{sim}(v_i, t_i)/\tau\right)}{\sum_{j=1}^{N} \exp\left(\mathrm{sim}(v_i, t_j)/\tau\right)}, \tag{4}$$

where $N$ is the batch size, $\mathrm{sim}(\cdot)$ denotes cosine similarity and $\tau$ is a temperature parameter. The overall pretraining loss is defined as:

$$\mathcal{L}_{pretrain} = \lambda_1 \mathcal{L}_{mlm} + \lambda_2 \mathcal{L}_{lm} + \lambda_3 \mathcal{L}_{seg} + \lambda_4 \mathcal{L}_{con}, \tag{5}$$

where $\lambda_1$, $\lambda_2$, $\lambda_3$, and $\lambda_4$ being empirically tuned weights. The joint optimization of these losses enables our model to capture both fine-grained local details and global semantic context. This robust initialization is pivotal for the subsequent Attribution Report Generation Stage, where further fine-tuning refines forgery localization and enhances the quality of the generated reports.

## 3.2 FORGERY PROMPTER NETWORK

**Motivation.** Accurately identifying the most salient manipulated regions in forged images is challenging due to the high visual fidelity of modern manipulation techniques. Even human reviewers must closely inspect the images to detect inconsistencies. Therefore, we propose the Forgery Prompter Network (FPN) to generate an initial set of salient region keywords, which guide downstream reasoning and facilitate the coherent generation of attribution reports.

**Region Keywords Extraction.** We extract region labels from the textual descriptions. The label space comprises 21 facial semantics, where each image is associated with a 21-dimensional vector $Y$; the $i$-th element is 1 if the corresponding facial part is mentioned in the textual description, and 0 otherwise.

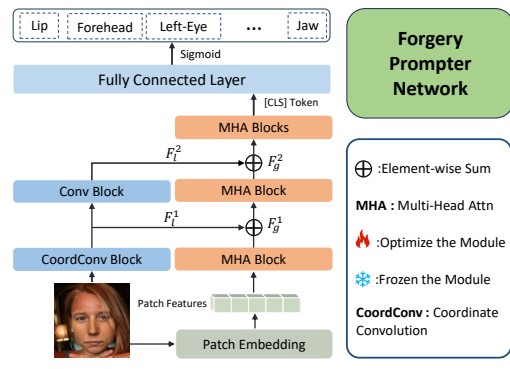

Figure 5: Illustration of the Forgery Prompter Network (FPN). The FPN generates region-aware prompts for forgery localization.

**Forgery Prompter Network (FPN)** takes the vision transformers as the main architecture. Considering the crucial role of fine-grained local context in identifying subtle flaws, we introduce a convolution branch at the early $m$ layers to complement the global contexts captured by the vision transformer. As shown in Figure 5, the forgery image $I$ concurrently traverses self-attention blocks and convolution blocks in parallel, producing global-aware features $F_g = \{F_g^0, F_g^2, ..., F_g^{m-1}\}$ and local-aware features $F_l = \{F_l^0, F_l^1, F_l^2, ..., F_l^{m-1}\}$. At each encoding level, the corresponding features are element-wise summed and fed into next attention block:

$$F_g^i = \mathrm{MHA}_{i-1}(F_g^{i-1}), F_l^i = \mathrm{Conv}_{i-1}(F_l^{i-1}), \tag{6}$$

$$F_g^i = \mathrm{MHA}_i(F_g^i + F_l^i), \quad i = 1, \cdots, m \tag{7}$$

where "MHA" and "Conv" mean the multi-head attention and convolution. Furthermore, we note that the positioning of facial regions in a natural image follows a rigid and predictable structure, with the eyes typically positioned laterally relative to the nose and the eyebrows aligned above the eyes. Leveraging this regularity, we integrate coordinate convolution (Liu et al., 2018) in the initial convolutional layer to detect anomalies in the arrangement of facial features, *i.e.,* $\text{Conv}_0 = \text{CoorConv}$. The resultant feature $F_g^m$ contains both global and local contexts and is fed into subsequent multi-head attention blocks and a classification head to produce the probability $\hat{Y}$ across regions, while also being used in cross-attention with Q-former features to enhance forgery localization. Finally, the forgery prompter network is trained using a combined loss, incorporating both Binary Cross-Entropy (BCE) and Dice loss to effectively balance region classification and overlap precision:

$$\mathcal{L}_{BCE} = -\frac{1}{21} \sum_{i=1}^{21} Y_i \log \hat{Y}_i + \omega(1 - Y_i) \log(1 - \hat{Y}_i), \tag{8}$$

where $\omega$ is a discount factor set to $\omega < 1$ to address the imbalance due to the prevalence of unmodified regions. The Dice loss is employed to measure the overlap between the predicted labels $\hat{Y}$ and ground truth $Y$, ensuring that less frequent classes receive more attention:

$$\mathcal{L}_{Dice} = 1 - \frac{2 \sum_{i=1}^{21} Y_i \hat{Y}_i}{\sum_{i=1}^{21} Y_i + \sum_{i=1}^{21} \hat{Y}_i}. \tag{9}$$

Finally, we optimize FPN with the average of the BCE and Dice losses via $\frac{1}{2}(\mathcal{L}_{BCE} + \mathcal{L}_{Dice})$.

### 3.3 ATTRIBUTION REPORT GENERATION

Subsequently, we fix the trained FPN network and take its region predictions as prior clues to aid both the report generation and the cross-attention process for improved forgery localization. Assume the set of regions from the FPN is $R = \{r_1, r_2, ...\}$. We design a particular template to include $R$ and form a report-focused instruction $\text{T}_{instr}$:

```
These facial areas may be manipulated by AI: [R]. Please describe
the specific issues in these areas.
```

This structured prompt serves as the guiding context for the language model, thereby ensuring that the final output accurately reflects the manipulations detected by the FPN. This integration enhances the coherence and quality of the generated reports, offering a comprehensive understanding of the tampered regions. Subsequently, the instruction and the image embeddings are fed into the Q-former, and the resulting features are passed to the large language model to generate the explanatory text $\hat{T}$ with the length of $L_{\hat{T}}$. This output is then supervised by the language modeling loss as:

$$\mathcal{L}_t = -\mathbb{E}_{(I,T) \sim \mathcal{D}} \left[ \sum_{k=1}^{L_{\hat{T}}} \log P\left( \hat{T}_k \mid I, \hat{T}_0, \ldots, \hat{T}_{k-1} \right) \right], \tag{10}$$

where $(I, T) \sim \mathcal{D}$ indicates that the expectation is taken over samples from the dataset $\mathcal{D}$.

### 3.4 MASK DECODER

We employ SAM's Two-way Transformer (Kirillov et al., 2023) as the mask decoder. The image encoder of InstructBLIP encodes the forgery image. The resulting features from the Q-former are then enhanced through cross-attention with the FPN's regional prompts. These enriched features are subsequently fed into the Two-way Transformer to predict the forgery mask $\hat{M}$. The cross-entropy loss is applied:

$$\mathcal{L}_m = -\frac{1}{HW} \sum_{i=1}^{H} \sum_{j=1}^{W} \left[ M_{ij} \log \hat{M}_{ij} + (1 - M_{ij}) \log(1 - \hat{M}_{ij}) \right], \tag{11}$$

where $H, W$ are the height and width of the image. Overall, the full loss in the second stage for report generation and forgery localization is formulated as $\mathcal{L}_{full} = \mathcal{L}_t + \mathcal{L}_m$.

Table 2: Performance comparison of generated captions and forgery localization across models. "Report Generation" metrics evaluate caption relevance and diversity, while "Forgery Localization" metrics assess accuracy in identifying tampered regions.

| Method | Reference | Report Generation | | | | | | | Forgery Localization | | |
|---|---|---|---|---|---|---|---|---|---|---|---|
| | | CIDEr | METEOR | BLEU-1 | BLEU-2 | BLEU-3 | BLEU-4 | ROUGE-L | IoU | Precision | Recall |
| SCA (Huang et al., 2024) | CVPR24 | 40.6 | 29.2 | 30.6 | 17.8 | 11.2 | 8.2 | 27.6 | 46.69 | 48.49 | **92.11** |
| LISA-7B (Lai et al., 2024) | ICCV23 | 44.1 | 28.1 | 31.1 | 17.9 | 10.8 | 8.5 | 28.4 | 52.45 | 73.26 | 71.53 |
| Osprey (Yuan et al., 2024) | CVPR24 | 24.5 | 27.5 | 28.7 | 16.4 | 9.4 | 6.2 | 25.9 | - | - | - |
| InstructBLIP (Dai et al., 2023) | NeurIPS23 | 51.7 | 14.6 | 31.8 | 20.3 | 14.6 | 11.4 | 27.7 | 64.04 | 87.88 | 78.53 |
| **ForgeryTalker** | - | **59.3** | 15.9 | **35.0** | **22.1** | **16.0** | **12.5** | **28.8** | **73.67** | **91.43** | 86.22 |

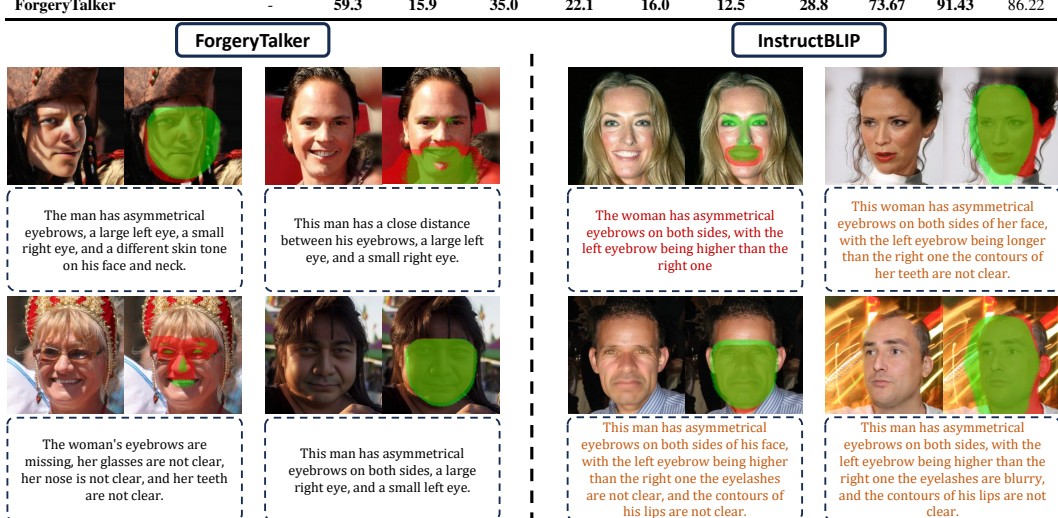

Figure 6: Qualitative comparison of ForgeryTalker and InstructBLIP. For the results, the predicted mask is shown in green and the ground-truth in red to highlight localization errors.

## 4 EXPERIMENT

In this section, we present a series of experiments to evaluate our proposed model, ForgeryTalker, on the MMTT dataset against several baselines.

### 4.1 QUANTITATIVE RESULTS

As shown in Table 2, we benchmark our proposed baseline, ForgeryTalker, against several existing models adapted for our task: SCA (Huang et al., 2024), LISA-7B (Lai et al., 2024), Osprey (Yuan et al., 2024), and InstructBLIP (Dai et al., 2023).

**Report Generation.** ForgeryTalker obtains the highest CIDEr score (59.3), surpassing InstructBLIP (51.7), LISA-7B (44.1), SCA (40.6), and Osprey (24.5). It also achieves the best performance across all BLEU scores, with a BLEU-1 of 35.0 and BLEU-4 of 12.5, and leads in ROUGE-L (28.8). The comparatively lower scores of SCA and Osprey suggest they produce more simplistic reports. To further verify the report generation of ForgeryTalker, we augment the evaluation to the DQ_F++ dataset (Zhang et al., 2024b). Results are reported in Table 3, our ForgeryTalker variants demonstrate superior generalization compared to the baselines. Specifically, our final configuration (2:1:1:1) achieves the highest CIDEr score of **113.6**, significantly outperforming the next best, InstructBLIP (98.5). Concurrently, our 1:1:1:2 configuration leads on all fluency-related metrics, including Bleu-1 (**48.5**) and ROUGE-L (**47.2**). These strong results validate that our method generalizes effectively to unseen datasets.

**Forgery Localization.** ForgeryTalker achieves the highest IoU (73.67) and Precision (91.43). Its competitive Recall (86.22) is second only to SCA, which achieves the highest Recall of 92.11 but with a notably lower IoU (46.69) and Precision (48.49). Other baselines like LISA-7B and Instruct-BLIP report IoUs of 52.45 and 64.04, respectively. Note that Osprey does not provide standalone forgery masks, so its localization metrics are not reported. The qualitative results in Figure 6 visually corroborate these findings. While InstructBLIP's predicted masks (green) often over-segment beyond the ground-truth (red) and its reports are verbose, ForgeryTalker consistently produces more precise masks and concise, relevant reports.

Table 3: Report generation comparison on the DQ_F++ dataset (Zhang et al., 2024b). The best score for each metric is shown in **bold**.

| Model | Bleu_1 | Bleu_2 | Bleu_3 | Bleu_4 | ROUGE_L | CIDEr |
|---|---|---|---|---|---|---|
| SCA | 47.6 | 39.9 | 35.2 | 30.1 | 40.4 | 71.0 |
| LISA | 46.5 | 38.4 | 33.1 | 31.2 | 45.6 | 74.3 |
| InstructBLIP | 43.5 | 38.0 | 34.2 | 31.0 | 47.9 | 98.5 |
| ForgeryTalker (2:1:1:1) | 47.1 | 40.0 | 35.1 | 31.0 | 46.8 | **113.6** |
| ForgeryTalker (1:1:1:2) | **48.5** | **41.4** | **36.5** | **32.4** | 47.2 | 113.3 |

Table 4: Ablation study on the impact of different variants. $w/$ and $w/o$ mean equipping or not equipping the following modules.

| Method | Report Generation | | | | | | | Forgery Localization | | |
|---|---|---|---|---|---|---|---|---|---|---|
| | CIDEr | METEOR | Bleu_1 | Bleu_2 | Bleu_3 | Bleu_4 | ROUGE_L | IoU | Precision | Recall |
| ForgeryTalker $w/$ FPN-GT | **95.1** | **20.6** | **41.5** | **27.6** | **20.3** | **16.0** | **37.0** | 66.90 | 88.74 | 79.83 |
| ForgeryTalker $w/o$ FPN | 51.7 | 14.6 | 31.8 | 20.3 | 14.6 | 11.4 | 27.7 | 64.04 | 87.88 | 78.53 |
| ForgeryTalker | 59.3 | 15.9 | 35.0 | 22.1 | 16.0 | 12.5 | 28.8 | **73.67** | **91.43** | **86.22** |

Table 5: Ablation study results for different pretraining loss weight settings $(\lambda_1{:}\lambda_2{:}\lambda_3{:}\lambda_4)$ as defined in Eq. 5. Here, $\lambda_1$, $\lambda_2$, $\lambda_3$, and $\lambda_4$ denote the weights for the masked language modeling, language modeling, segmentation, and contrastive losses.

| Loss Ratio $(\lambda_1{:}\lambda_2{:}\lambda_3{:}\lambda_4)$ | Report Generation | | | | | | Forgery Localization | | |
|---|---|---|---|---|---|---|---|---|---|
| | CIDEr | BLEU-1 | BLEU-2 | BLEU-3 | BLEU-4 | ROUGE-L | IoU | Precision | Recall |
| $w/o$ Pretraining Stage | 54.4 | 33.8 | 21.6 | 15.5 | 12.1 | 28.3 | 65.87 | 89.00 | 78.87 |
| 1:1:1:1 | 57.1 | 35.1 | 21.9 | 15.7 | 12.2 | 28.9 | 72.33 | 90.90 | 86.42 |
| 1:1:1:2 | 59.0 | 33.5 | 20.9 | 15.2 | 12.0 | 28.5 | 74.04 | 91.34 | 86.74 |
| 1:1:2:1 | 57.6 | 34.1 | 21.4 | 15.4 | 12.1 | 28.6 | 73.56 | 91.44 | 86.16 |
| 1:2:1:1 | 57.9 | 30.3 | 19.1 | 14.1 | 11.3 | 27.8 | **73.92** | 91.05 | 86.47 |
| **2:1:1:1** | **59.3** | **35.0** | **22.1** | **16.0** | **12.5** | **28.8** | 73.67 | 91.43 | 86.22 |

## 4.2 ABLATION STUDY

**Effect of the Forgery Prompter Network (FPN).** The FPN is shown to be a critical component. Table 4 reveals a significant performance drop in report generation (CIDEr drops to 51.7) when the FPN is removed. Conversely, an oracle FPN using ground-truth prompts (w/ FPN-GT) establishes a high upper bound at 95.1 CIDEr. This large performance gap underscores that the quality of region prompts is a key factor for this task and motivates future work on improving the FPN module.

**Pretraining Stage.** Our forgery-aware pretraining stage proves highly effective, substantially boosting both localization and report generation over a baseline without it (*e.g.*, IoU +8.8, CIDEr +4.9), as shown in Table 5. Among the tested loss weight ratios, we selected 2:1:1:1 $(\lambda_1{:}\lambda_2{:}\lambda_3{:}\lambda_4)$ as it provides the best overall trade-off. It achieves the highest report generation scores (59.3 CIDEr) while maintaining a competitive localization performance (73.67 IoU), making it the optimal choice for our joint task.

## 5 CONCLUSION

In this paper, we address the limitations of traditional forgery localization methods, which typically provide only binary masks and lack sufficient explanatory power. To move beyond this, we introduce and formalize the novel task of Forgery Attribution Report Generation, aiming to produce both precise localization masks and rich, human-readable textual explanations. To catalyze research in this new direction, we construct and release the **MMTT** dataset, the first large-scale benchmark for this task, featuring high-precision, programmatically-generated masks and meticulously crafted textual annotations. Furthermore, we propose **ForgeryTalker**, a powerful baseline model that effectively unifies the localization and report generation processes into a single end-to-end framework. Our comprehensive experiments not only validate the effectiveness of ForgeryTalker but also establish a solid benchmark on the MMTT dataset. We believe our contributions, the new task, the public dataset, and the effective baseline, will pave the way for future advancements in explainable and trustworthy facial forgery analysis.

## ETHICS STATEMENT

We affirm that this work complies with the ICLR Code of Ethics. The MMTT dataset and corresponding analyses were developed exclusively for research on localizing and explaining the forgery of facial images. We acknowledge the dual-use nature of synthesizing realistic and semantically consistent examples—the methods involved could potentially be repurposed to create deceptive content. To mitigate such risks, we implement a controlled-release protocol: (i) the full generation pipeline, detailed prompts, and prompt–response pairs will not be disclosed to prevent malicious use; (ii) public access to the dataset will be restricted to research purposes under a Data Usage Agreement (DUA); (iii) high-resolution originals and sensitive metadata will be retained securely; (iv) content involving minors and explicitly sensitive real-world conflict scenarios has been excluded; and (v) access may be revoked in cases of misuse.

## REPRODUCIBILITY STATEMENT

We are committed to research reproducibility. To support this, we would release code, experiments configuration details, pretrained checkpoints, and our MMTT dataset to reproduce all results reported in this paper. (1) Code: Our source code, including model implementations, training/evaluation scripts, and analysis tools, will be publicly released on GitHub. The repository includes a detailed README.md with setup instructions to facilitate replication of our results. (2) Data: The dataset will be made accessible to academic researchers under a Data Use Agreement. (3) Experimental Setup: Detailed hyperparameters, software/hardware environment specifications, and metric computation procedures are provided in Appendix.

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

## A    RELATED WORK

**Facial Manipulation Localization.** Detecting manipulated facial regions, especially deepfakes, has garnered attention. CNN-based methods (Sabir et al., 2019) utilize temporal inconsistencies for videos, while GAN-based approaches, such as GANprintR (Neves et al., 2020) and MaskGAN (Liu et al., 2022a), address synthetic artifacts. Hybrid models like HCiT (Kaddar et al., 2021) combine CNNs and ViTs to enhance generalization, and multi-modal methods (Sun et al., 2023; Khedkar et al., 2022) leverage spatial-temporal inconsistencies. However, these models lack interpretability and fine-grained mask generation, which our work addresses by providing both localization masks and textual explanations.

**Multi-label Classification for Facial Localization.** Multi-label classification captures independent alterations in facial regions but struggles with dependencies across features. CNNs (Lalitha & Sooda, 2022) face limitations in fine-grained tasks, while hybrid models (Kaddar et al., 2021) improve detection by combining local and global features. Weighted loss functions (Ramachandran et al., 2021) and parallel branches (Richards et al., 2023) address class imbalance and refine detection. Yet, few works integrate multi-label classification with localization. Our ViT-based classifier bridges this gap by capturing complex dependencies with parallel branches and weighted loss functions.

**Segmentation Techniques.** Segmentation is crucial for identifying localized manipulations. Models like U-Net and DeepLab (Ross & Dollár, 2017) focus on spatial features, while Transformer models (Alexey, 2020) capture global context. Recent methods like SAM (Kirillov et al., 2023) use a Two-Way Transformer for high-quality masks but lack manipulation-specific context. By integrating SAM with InstructBLIP, we create context-aware forgery masks, unifying segmentation and manipulation detection for enhanced localization.

**Explainable Forgery Detection.** A recent trend is moving towards explainable forgery detection, with FakeShield (Xu et al., 2025) being the most closely related work. While FakeShield addresses general image forgery and constructs its dataset using GPT-4o, our work introduces MMTT, the first benchmark focusing specifically on the facial forgery domain with meticulous human annotation.

## B    EXAMPLES FROM MMTT DATASET

To enhance the understanding of the MMTT dataset and its unique contributions to facial image forgery localization, we provide a word cloud generated from the textual descriptions (captions) and a series of representative examples. The MMTT dataset is meticulously designed to facilitate fine-grained forgery localization by leveraging multimodal annotations. Each sample consists of three complementary components: a manipulated image, a binary mask delineating the forged regions, and a detailed textual description that explicitly identifies and contextualizes the alterations. These comprehensive annotations provide a robust foundation for research tasks requiring precise localization and explainability of facial manipulations.

The word cloud, presented in Figure 8, visually encapsulates the linguistic distribution within the dataset's textual annotations. Dominant terms such as "woman," "man," "skin tone," and "facial skin" highlight the dataset's focus on describing forgery in specific facial regions. Furthermore, frequent mentions of region-specific features, such as "left eye," "nose bridge," and "right eyebrow," underscore the granularity and specificity of the annotations. This visualization demonstrates the alignment between the textual descriptions and the underlying task of forgery localization, offering an overview of the dataset's descriptive richness and consistency.

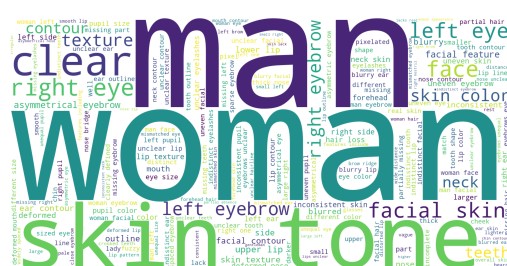

Figure 8: Word cloud of captions in the MMTT dataset. This visualization highlights the most frequently used words in the dataset's textual descriptions, where the font size represents the frequency of occurrence.

Figure 7 illustrates selected examples from the MMTT dataset, showcasing its multimodal struc-

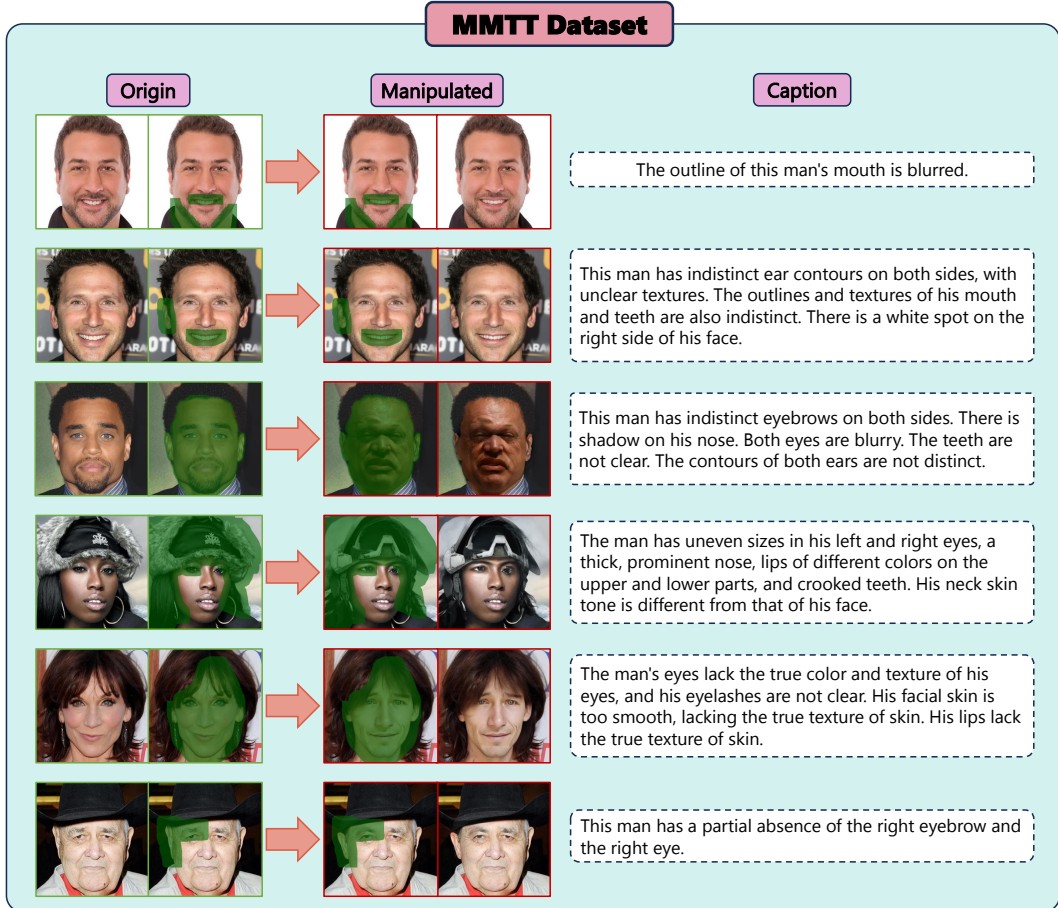

Figure 7: Examples from the MMTT dataset. Each row illustrates a case from the dataset, comprising a manipulated image, its corresponding binary mask (overlaid in green), and a textual description detailing the altered facial regions. For illustrative purposes, the original (authentic) images are also included in this figure to highlight the extent and nature of the manipulations. The green regions indicate the localized areas of forgery as identified by the binary masks. It is important to note that the original images are not part of the MMTT dataset; the dataset itself consists only of manipulated images, binary masks, and their associated textual descriptions.

ture and the diversity of forgery types. Each example includes a manipulated image, its corresponding binary mask, and a textual description. For illustrative purposes, we have also included the original (authentic) images alongside the manipulated samples in Figure 7 to provide additional context for understanding the nature and extent of the forgeries. It is important to note that these original images are not part of the MMTT dataset and are shown exclusively to highlight the transformations and to provide clarity on the dataset's structure. The actual dataset is focused on forged images, binary masks, and detailed captions, without the inclusion of original (authentic) images.

## C    EXPERIMENTAL SETUP

We implement ForgeryTalker with PyTorch (Paszke et al., 2019) and train on four NVIDIA A100 GPUs, using an 8:1:1 train/validation/test split of the MMTT dataset. The two-stage training process is as follows: (1) The FPN is trained for 125k steps (batch size 16, initial lr 7.5e-3 with cosine decay) with the BCE loss weight $\omega$ set to 0.2. (2) With the FPN frozen, the main model is trained for 60 epochs (batch size 16, lr 4e-6) using mixed-precision (fp16) training.

Table 6: Performance comparison against state-of-the-art models in the supplementary material. Specialist models (IML-ViT, PSCC-NET) and our ForgeryTalker are trained on our dataset. **General Large Vision-Language Models (LVLMs) are evaluated in a zero-shot setting.** Best performance for each metric is in **bold**.

| Model | Interpretation Generation | | | | | | Forgery Localization | | |
|---|---|---|---|---|---|---|---|---|---|
| | CIDEr | ROUGE_L | Bleu_1 | Bleu_2 | Bleu_3 | Bleu_4 | IoU | Precision | Recall |
| Seed1.5VL (Guo et al., 2025) | 0.54 | 13.74 | 17.19 | 5.35 | 1.86 | 0.98 | - | - | - |
| Qwen2.5VL (72B) (Bai et al., 2025) | 2.72 | 16.52 | 20.34 | 6.33 | 2.55 | 1.46 | - | - | - |
| Qwen2.5VL (32B) (Bai et al., 2025) | 2.53 | 16.89 | 22.44 | 8.16 | 3.3 | 1.78 | - | - | - |
| Qwen2.5VL (7B) (Bai et al., 2025) | 2.48 | 17.0 | 22.33 | 8.2 | 3.24 | 1.78 | - | - | - |
| llava (72B) (Liu et al., 2024) | 3.06 | 16.83 | 22.01 | 8.35 | 3.3 | 1.8 | - | - | - |
| llava (8B) (Liu et al., 2024) | 2.1 | 18.05 | 20.75 | 7.66 | 3.15 | 1.75 | - | - | - |
| InternVL3 (78B) (Zhu et al., 2025a) | 2.67 | 16.82 | 22.31 | 8.08 | 3.16 | 1.75 | - | - | - |
| InternVL3 (38B) (Zhu et al., 2025a) | 2.29 | 17.18 | 21.21 | 7.83 | 3.2 | 1.76 | - | - | - |
| InternVL3 (14B) (Zhu et al., 2025a) | 2.29 | 16.85 | 20.81 | 7.54 | 3.03 | 1.71 | - | - | - |
| IML-ViT (Ma et al., 2023) | - | - | - | - | - | - | **77.89** | 83.76 | **90.04** |
| PSCC-NET (Liu et al., 2022b) | - | - | - | - | - | - | 32.33 | 70.3 | 37.44 |
| **ForgeryTalker** | **59.3** | **28.8** | **35.0** | **22.1** | **16.0** | **12.5** | 73.67 | **91.43** | 86.22 |

# D COMPARISON WITH STATE-OF-THE-ART MODELS

We compare the performance of our proposed ForgeryTalker framework against a range of recent models, including specialist forgery localization models and general-purpose Large Vision-Language Models (LVLMs). It is important to note that while the specialist models and our ForgeryTalker were trained on our dataset, the general LVLMs were evaluated in a zero-shot setting to assess their out-of-the-box capabilities for this novel task. The evaluation covers both forgery localization (IoU, Precision, Recall) and interpretation generation (CIDEr, ROUGE-L, BLEU scores), with results summarized in Table 6.

For forgery localization, specialist models like IML-ViT expectedly achieve the highest scores in IoU (77.89) and Recall (90.04), as they are solely optimized for this task. However, our ForgeryTalker demonstrates highly competitive localization capabilities, achieving a strong IoU of **73.67** and securing the best Precision score (**91.43**) among all compared models. This indicates our model's superior ability to avoid over-predicting forged regions.

For the primary task of interpretation generation, ForgeryTalker significantly outperforms all other LVLMs across every text-based metric. It achieves a CIDEr score of **59.3**, which is an order of magnitude higher than the next best competitor, Llava-72B (3.06). This substantial gap highlights the effectiveness of our forgery-aware architecture in generating accurate and relevant textual explanations, a task where general-purpose LVLMs, which were not fine-tuned on our forgery-specific data, naturally struggle. In summary, ForgeryTalker establishes a new state-of-the-art in interpretable forgery localization by providing best-in-class captioning performance while maintaining a robust and precise localization ability.

# E ABLATION STUDY ON THE IMPACT OF THE FPN

**Impact of FPN Loss and Discount Factor.** We use Positive Label Matching (PLM) to evaluate the effectiveness of FPN. PLM calculates the ratio of correctly predicted positive labels over the union of predicted and ground-truth positive labels:

$$\text{PLM} = \frac{|\text{Predicted Positive Labels} \cap \text{Ground Truth Positive Labels}|}{|\text{Predicted Positive Labels} \cup \text{Ground Truth Positive Labels}|}. \tag{12}$$

Unlike IoU, PLM focuses on detecting manipulated regions without being influenced by a large number of correctly predicted negative labels, making it ideal for tasks with sparse modifications.

The forgery prompter network is optimized by a combined loss, incorporating both Binary

| Model | $\omega$ | Loss | PLM |
|---|---|---|---|
| ViT | 1 | BCE | 34.23 |
| ViT | 0.2 | BCE | 38.92 |
| FPN | 0.2 | BCE | 39.16 |
| **FPN** | 0.2 | BCE + Dice | **41.05** |

Table 7: Ablation Study on the Impact of the FPN

Cross-Entropy (BCE) loss and Dice loss to effectively balance region classification and overlap precision:

$$\mathcal{L}_{BCE} = -\frac{1}{21} \sum_{i=1}^{21} Y_i \log \hat{Y}_i + \omega(1 - Y_i) \log(1 - \hat{Y}_i), \tag{13}$$

where $\omega$ is a discount factor set to $\omega < 1$ to address the imbalance due to the prevalence of unmodified regions.

Table 7 examines the effect of the discount factor $\omega$ in the BCE loss (Eq. 13) and the addition of Dice loss. Setting $\omega = 0.2$ improves the PLM metric from 34.23 to 38.92 on a ViT backbone; further incorporating the FPN boosts PLM to 39.16, and combining BCE with Dice raises it to 41.05. This confirms that discounting unmodified regions and combining losses enhances region prompt accuracy.

## F  LLM USAGE STATEMENT

During the preparation of this work, the authors used a large language model to assist with improving grammar, rephrasing sentences, and ensuring terminological consistency. The authors reviewed and edited all model-generated text and take full responsibility for the final content of this paper.

