# OpenReview forum: "Where and Why in Image Forgery: A Benchmark for Joint Localization and Explanation"
_ICLR.cc/2026/Conference — ICLR 2026 Conference Withdrawn Submission_

### Official Review · Reviewer_DR5r · 2025-10-24

**Soundness:** 3
**Presentation:** 2
**Contribution:** 3
**Rating:** 4
**Confidence:** 4

**Summary:**

This paper introduces Forgery Attribution Report Generation, a new forgery task that simultaneously localizes forged image regions (“Where”) and generates human-readable explanations of the manipulations (“Why”). To address the challenge, the authors release Multi-Modal Tamper Tracing (MMTT), a large-scale dataset with human annotation.

The authors further propose ForgeryTalker, a unified vision–language framework for forgery localization and textual explanation. The model is trained in two stages for both alignments and finetuning. Extensive experiments show that ForgeryTalker outperforms strong language and localization baselines (e.g., InstructBLIP, LISA-7B, SCA) in both textual report generation and forgery localization. Ablation studies confirm the importance of the Forgery Prompter Network (FPN) and the pretraining design.

Overall, the work contributes a new benchmark, task definition, and baseline model for explainable image forgery analysis, moving beyond binary classification toward interpretive forensics.

**Strengths:**

1. The paper clearly formalizes localization and explanation as a joint problem for image forensics to provide interpretability.

2. The proposed MMTT is a large-scale (150K+) dataset that integrates pixel-level masks with linguistical annotations.

3. ForgeryTalker elegantly integrates multimodal reasoning with a shared encoder and dual decoders, enabling joint optimization on localization and text explanation.

4. The model achieves state-of-the-art performance across both language (CIDEr and BLEU) and localization (IoU, Precision) metrics, outperforming existing baselines.

**Weaknesses:**

Major Weaknesses:

1. While the paper proposes the new task of joint localization and explanation, the necessity for combining these subtasks is not clearly justified. Given that both have been studied in the field of image forgery, the authors should clarify how their joint formulation yields additional insights or mutual improvements.

2. In Fig. 4, the textual instruction fed to the mask decoder differs in Forgery-aware Pretraining Stage and Forgery Generation Stage, which may introduce potential biases.

3. The baselines are limited to general multimodal language models, while existing multimodal forgery explanation methods (e.g., FakeShield[1], M2F2-Det[2]) are absent.

4. The method is evaluated only on DQ F++, another face-forgery dataset. Broader out-of-distribution forgeries would strengthen the authors' claims.

Minor Weaknesses:

1. Some inconsistent names and definitions (e.g., “Forgery Attribution Report Generation” vs. “Joint Forgery Localization and Explanation”, "Forgery Generation Stage" vs. "Attribution Report Generation Stage") may confuse readers.

2. Typo in "Cross-model Alignment Learning". "Cross-model" should be "Cross-modal".

[1] Xu, Zhipei, et al. "Fakeshield: Explainable image forgery detection and localization via multi-modal large language models." arXiv preprint arXiv:2410.02761 (2024).
[2] Guo, Xiao, et al. "Rethinking Vision-Language Model in Face Forensics: Multi-Modal Interpretable Forged Face Detector." Proceedings of the Computer Vision and Pattern Recognition Conference. 2025.

**Questions:**

1. What advantages arise from jointly forgery localization and explanation compared to handling them separately?

2. Details of the textual instructions fed to the mask decoder between pretraining and generation stages are expected.

3. The authors should include multimodal forgery explanation methods (e.g., FakeShield, M2F2-Det) for fair comparisons.

4. How does the model generalize to out-of-distribution forgeries in other benchmarks?

**Details Of Ethics Concerns:**

The authors explicitly recognize dual-use concerns and include a responsible data-release policy.

---

> ### Author Response · Authors · 2025-11-14
> **Response to Reviewer DR5r**
>
> We thank Reviewer DR5r for their review. We appreciate that you recognized the strengths of our work, such as the clear formalization of the joint problem, the value of the MMTT dataset, and the design of the ForgeryTalker framework.
>
> You have raised several major weaknesses that we agree are critical to our paper.
>
> Your first point regarding the necessity of the joint task is a fundamental question. We acknowledge that we failed to provide a clear justification for _why_ joint localization and explanation are mutually beneficial, compared to addressing them as separate tasks.
>
> We also agree that our baselines are incomplete. We are missing crucial comparisons to recent multimodal forgery explanation methods (such as FakeShield and M2F2-Det), which is a significant omission.
>
> Furthermore, we take your points about the limited evaluation (lacking broader out-of-distribution tests) and the potential for bias from inconsistent textual instructions during training. These are valid concerns that we did not adequately address.
>
> We will carefully consider these points, as well as the minor terminological issues you noted, in our future work. Thank you for this constructive feedback.

---

### Official Review · Reviewer_tKNC · 2025-10-24

**Soundness:** 3
**Presentation:** 3
**Contribution:** 3
**Rating:** 4
**Confidence:** 4

**Summary:**

The paper introduces a new large-scale dataset of real images from FFHQ and CelebA, manipulated using three groups of techniques (namely, face swapping, face editing, and image inpainting). The generated deepfakes are then shown to 30 trained human annotators, who, after observing the images and manipulation masks for a minimum of 1 minute, write a text description of the obvious areas of manipulation.
Consequently, the dataset is used to train an MLLM on mask prediction and interpretation. The MLLM extends the InstructBLIP architecture by incorporating a SAM decoder and several losses to improve performance.

**Strengths:**

- The paper is well written and easy to follow. The steps to obtain the manipulated data and annotations can be replicated from the paper and supplementary material. Similarly, the MLLM architecture is relatively clear (although it has many components).
- In terms of performance, ForgeryTalk performs better compared to InstructBlip on localisation which is expected. We also see good performance on DD_F++.

**Weaknesses:**

- Most weaknesses identified are in relation to the trained MLLM. First and foremost, the ablations seem incomplete. The method uses several losses and a two-stage training method. The ablations are limited to w/wo FPN and very few variations for the weight of each loss component; what happens if we only train end-to-end? what about if we drop the mlm loss?
- The model is trained on localisation, but this is assuming FPN has already detected the areas. Converting the class labels to coherent sentences is a trivial task for the MLLM, so the effectiveness of the architecture is a little misleading. Of course, the pipeline still works, but it seems overkill --i.e. why convert a set of class labels to sentences if they are already human-readable? The architecture would make more sense in a VQA format.
- The architecture seems to be trained only on forgeries. As such, we should expect some hallucinations and false positives on real images. This is, in fact, a drawback, as it assumes another pre-trained method ahead of the FPN to do the binary detection, further raising computational requirements.

Some minor points:
- It is unclear if the "augmented evaluation to DQ_F++" means cross-dataset generalisation or if the model if finetuned on the dataset
- line 281: there should be a new paragraph for this component, to maintain template consistency
- The mlm loss section is a little unclear. t is never defined, and it is also unclear what is the ground truth and reasoning behind it.

**Questions:**

- How does the model perform on other standard datasets, eg FF++?
- How does the model perform when contrastive and mlm losses are not used?
- How does the model behave when real images are given?
- Is table 3 showing cross-dataset generalisation?
- Why is the binary detection dropped? What is the motivation for such a choice?

---

> ### Author Response · Authors · 2025-11-14
> **Response to Reviewer tKNC**
>
> We thank Reviewer tKNC for their review and for finding the paper well-written, easy to follow, and the architecture clear. We appreciate the "good" ratings on soundness, presentation, and contribution.
>
> You have raised several significant weaknesses regarding the MLLM design and evaluation, which we find insightful.
>
> We agree that our ablation studies are incomplete. We are missing key analyses on alternative training strategies (like end-to-end) and the specific contributions of individual loss components (like the mlm and contrastive losses).
>
> Your criticism of the model's practical limitations is a critical point. We acknowledge that training only on forgeries and dropping binary detection is a major drawback. This design choice means the model would likely perform poorly (e.g., hallucinate) on real images, which is a significant issue we failed to address.
>
> We also take your point that the architecture may be 'overkill' for the task, especially if the FPN handles most of the localization, making the MLLM's contribution seem trivial. This is a valid concern about our design's effectiveness.
>
> Furthermore, we agree that our evaluation needs to be broadened to other standard datasets and that we must provide clearer experimental details (e.g., clarifying Table 3).
>
> These are important points that we will take into serious consideration for future revisions of this work. Thank you for your constructive feedback.

---

### Official Review · Reviewer_tBqx · 2025-10-26

**Soundness:** 2
**Presentation:** 2
**Contribution:** 2
**Rating:** 2
**Confidence:** 5

**Summary:**

The paper introduces a joint image forgery understanding task that requires models to determine both where an image has been manipulated and why. To support this, the authors build a new large-scale dataset (MMTT) containing over 150k manipulated images with pixel-level masks and human-written explanations covering multiple manipulation types (face swapping, attribute editing, inpainting, etc.). They also propose a unified model called ForgeryTalker.

**Strengths:**

1. The joint task of localization and explanation is clearly defined and offers a more interpretable understanding of image forgeries.
2. The proposed MMTT dataset contains over 150k samples with pixel-level masks and region-grounded textual descriptions. It covers diverse manipulation types and provides fine-grained part-level statistics.

**Weaknesses:**

1. If the dataset focuses primarily on face manipulations rather than general image edits, this should be explicitly stated in the title and abstract. Moreover, there are existing works on general image tampering that include both facial and non-facial examples with region-level and textual annotations, but comparisons and cross-domain evaluations with such datasets are missing.
2. Cross-dataset evaluation is very limited. There is no mask-level generalization test on established third-party localization datasets.
3. VLMs are evaluated in zero-shot mode, while ForgeryTalker is trained on the MMTT dataset. More importantly, comparisons with specialized deepfake or forgery detection methods are absent, making it difficult to assess whether the gains come from model design or data familiarity.
4. The paper claims over 150k human-written explanations, which implies a massive annotation effort. However, it does not specify how many annotators participated, how annotation consistency was ensured, or how quality control was performed. These details are crucial to assess dataset reliability for the benchmark.

**Questions:**

Issues raised in the *Weaknesses* section.

---

> ### Author Response · Authors · 2025-11-14
> **Response to Reviewer tBqx**
>
> We thank Reviewer tBqx for their highly confident and critical review of our submission. We appreciate that you recognized the clear definition of our proposed joint task and the attributes of the MMTT dataset.
> You have raised several fundamental concerns about our work, which we agree are significant and valid.
> We acknowledge that our evaluation methodology is limited. The baseline comparison (zero-shot VLM vs. our trained model) is not a fair assessment, and we are missing crucial comparisons to specialized deepfake and forgery detection methods. We also agree that our cross-dataset evaluation is insufficient and lacks generalization tests on established localization datasets.
> Furthermore, we recognize the critical points raised about our dataset. We agree that the scope requires clarification (face-focused vs. general image tampering) and that we failed to provide the necessary details about our annotation process (annotator count, consistency checks, quality control). These details are essential for assessing the benchmark's reliability.
> These are serious issues that we will need to address in future work. We thank you for your expertise in pointing out these fundamental flaws.

---

### Official Review · Reviewer_hdGe · 2025-10-29

**Soundness:** 3
**Presentation:** 3
**Contribution:** 3
**Rating:** 4
**Confidence:** 5

**Summary:**

This paper introduces a multimodal task of joint forgery localization and explanation, proposing the MMTT dataset with 152,217 samples containing forged images, pixel-level masks, and human-authored textual descriptions. The authors present ForgeryTalker, a unified framework that combines InstructBLIP with a Forgery Prompter Network (FPN) to generate both segmentation masks and natural language explanations of facial manipulations.

**Strengths:**

1. The paper constructs a large-scale dataset of 152,217 samples with careful human annotation, including both pixel-level masks and detailed textual descriptions, which will benefit the research community.
2. The paper successfully combines InstructBLIP, SAM's decoder, and a custom FPN to jointly address forgery localization and explanation generation.
3. The paper is easy to follow with informative figures and accessible writing that clearly motivates each design choice.

**Weaknesses:**

1. The annotation process may introduce confirmation bias since annotators are shown the ground-truth masks before writing descriptions. This guidance could lead them to describe artifacts that may not be perceptually obvious or even non-existent, especially for high-quality forgeries. The paper lacks inter-annotator agreement analysis or blind validation to verify annotation quality.

2. The paper only evaluates on MMTT (their own) and DQ_F++ datasets, both synthetic and research-oriented. Critical tests on real-world deepfakes, cross-forgery-tool generalization, and robustness to different manipulation methods are missing, raising concerns about practical applicability.

3. The paper does not explain why Q-former is necessary over the now-standard image encoder + LLM architecture. An ablation comparing direct ViT features + LLM versus the Q-former approach would help justify this design choice.

4. The paper lacks discussion and comparison with recent relevant works in multimodal face forgery detection, particularly: "FFAA: Multimodal Large Language Model Based Explainable Open-World Face Forgery Analysis Assistant", "MFCLIP: Multi-modal Fine-grained CLIP for Generalizable Diffusion Face Forgery Detection", "FakeShield: Explainable Image Forgery Detection and Localization via Multi-modal Large Language Models" (ICLR 2025), "Rethinking Vision-Language Model in Face Forensics: Multi-Modal Interpretable Forged Face Detector(CVPR2025)" and "Towards General Visual-Linguistic Face Forgery Detection" (CVPR 2025, which similarly uses mask information to assist annotation). Direct comparisons would better position this work's contributions.

5. While the paper ablates the FPN component, it lacks individual analysis of the four pretraining losses (Lmlm, Llm, Lseg, Lcon). How were the loss weights (2:1:1:1) determined? What happens if individual losses are removed? Table 5 shows only different weight combinations but no systematic analysis of each loss function's contribution.

**Questions:**

1. How do you ensure annotation quality given the confirmation bias risk? Since annotators are shown ground-truth masks before describing forgeries, what measures prevent them from "over-interpreting" high-quality fakes or describing non-existent artifacts? Did you conduct any blind validation or inter-annotator agreement tests?

2. Why does the FPN achieve only marginal improvement (39.16 vs 38.92 PLM)? Given the large performance gap between using ground-truth prompts (95.1 CIDEr) and predicted prompts (59.3 CIDEr), how do you address this bottleneck? Have you explored end-to-end training or alternative prompt generation strategies?

3. What are the computational costs and inference efficiency? How long does the two-stage training take, and what are the GPU memory requirements? What is the inference time per image? This is important for assessing the practical deployment feasibility of ForgeryTalker.

---

> ### Author Response · Authors · 2025-11-14
> **Response to Reviewer hdGe**
>
> We thank Reviewer hdGe very much for their detailed and insightful review of our submission. We sincerely appreciate the positive feedback on the value of the MMTT dataset, our model's framework combination, and the clarity of the paper. You have raised several critical and valid points that highlight significant shortcomings in our current work. We agree that these are important issues that need to be addressed.
>
> Specifically, we recognize the following limitations you pointed out:
> - The potential for confirmation bias in our annotation process and the need for a more rigorous analysis of annotation quality, such as inter-annotator agreement or blind validation.
> - The limited scope of our evaluation, which lacks testing on real-world deepfakes and analysis of generalization across different forgery tools and methods.
> - The lack of justification for key architectural choices, such as the necessity of the Q-former over simpler alternatives.
> - The missing comparisons to several highly relevant and recent works in multimodal forgery detection (like FakeShield, etc.), which are crucial for positioning our contribution.
> - The incompleteness of our ablation studies, particularly regarding a systematic analysis of the individual pretraining loss components and their weights.
> - The failure to address the FPN performance bottleneck (the gap between predicted and ground-truth prompts) and the omission of critical computational costs and inference efficiency metrics.
>
> Your review provides very clear and valuable guidance for us to improve this research in our future work. We will seriously consider and digest these insightful comments as we move forward. Thank you again for your valuable time and expertise.

---

### Note · Authors · 2025-11-14

**Comment:**

We would like to sincerely thank the Area Chair and all reviewers (hdGe, tBqx, tKNC, DR5r) for their time and effort in providing detailed and constructive feedback on our submission (#1894).

The reviews have raised several critical and valid concerns, particularly regarding our evaluation methodology, the need for more rigorous dataset validation, insufficient baseline comparisons, and the justification for key design choices.

We agree that these significant issues cannot be adequately addressed within the author discussion period. Therefore, we have decided to withdraw our submission to thoroughly revise and substantially improve the manuscript based on this valuable feedback.

We are grateful for the guidance provided by the review process.

**Withdrawal Confirmation:**

I have read and agree with the venue's withdrawal policy on behalf of myself and my co-authors.